# A Nuclear Factor Y-B Transcription Factor, *GmNFYB17*, Regulates Resistance to Drought Stress in Soybean

**DOI:** 10.3390/ijms23137242

**Published:** 2022-06-29

**Authors:** Maolin Sun, Yue Li, Jiqiang Zheng, Depeng Wu, Chunxia Li, Zeyang Li, Ziwei Zang, Yanzheng Zhang, Qingwei Fang, Wenbin Li, Yingpeng Han, Xue Zhao, Yongguang Li

**Affiliations:** Key Laboratory of Soybean Biology, Chinese Education Ministry, Northeast Agricultural University, Harbin 150030, China; smaolin2022@163.com (M.S.); lywcc19990804@163.com (Y.L.); zhengsichen1226@163.com (J.Z.); wdp90125@126.com (D.W.); 18747553072@163.com (C.L.); lzy780058207@163.com (Z.L.); dadouketi@126.com (Z.Z.); yanzhengzhang1992@163.com (Y.Z.); fqw2536901069@163.com (Q.F.); wenbinli@neau.edu.cn (W.L.); xuezhao@neau.edu.cn (X.Z.)

**Keywords:** GWAS, QTL, soybean, nuclear factor-Y, drought tolerance

## Abstract

Soybean is sensitive to drought stress, and increasing tolerance to drought stresses is an important target for improving the performance of soybean in the field. The genetic mechanisms underlying soybean’s drought tolerance remain largely unknown. Via a genome-wide association study (GWAS) combined with linkage analysis, we identified 11 single-nucleotide polymorphisms (SNPs) and 22 quantitative trait locus (QTLs) that are significantly associated with soybean drought tolerance. One of these loci, namely qGI10-1, was co-located by GWAS and linkage mapping. The two intervals of qGI10-1 were differentiated between wild and cultivated soybean. A nuclear factor Y transcription factor, *GmNFYB17,* was located in one of the differentiated regions of qGI10-1 and thus selected as a candidate gene for further analyses. The analysis of 29 homologous genes of *GmNFYB17* in soybean showed that most of the genes from this family were involved in drought stress. The over-expression of *GmNFYB17* in soybean enhanced drought resistance and yield accumulation. The transgenic plants grew better than control under limited water conditions and showed a lower degree of leaf damage and MDA content but higher RWC, SOD activity and proline content compared with control. Moreover, the transgenic plants showed a fast-growing root system, especially regarding a higher root–top ratio and more branching roots and lateral roots. The better agronomic traits of yield were also found in *GmNFYB17* transgenic plants. Thus, the *GmNFYB17* gene was proven to positively regulate drought stress resistance and modulate root growth in soybean. These results provide important insights into the molecular mechanisms underlying drought tolerance in soybean.

## 1. Introduction

Soybean is the world’s leading economic oil seed crop and provides quality protein and oil for human and animal consumption [1]. Soybean also provides biofuel production and many other products owing to its high protein and edible oil content [2]. Soybean is grown on an estimated 6% of the world’s arable land. Global soybean production was 17 million metric tons in 1960 but reached 363 million metric tons in 2019. The recent increase in production clearly shows the increased demand for soybean oil and high-protein meal.

Worldwide agricultural production has been limited by several environmental constraints in the form of abiotic stresses [3,4]. Among the abiotic stress factors, water deficitdramatically limits growth and yield in crops, particularly soybean, and the problem will likely be exacerbated by climate change [5]. Soybean has been estimated to have a 40% reduction in yield as a result of drought [6]. To overcome the negative impacts of drought stress in soybean, many strategies have been developed and adopted, mainly including agricultural practices and genetic improvement of soybean cultivars [7]. Rainfall and irrigation water is being used more efficiently; however, the adaptation of irrigation systems is region limited and would substantially increase the costs of soybean production [8]. Therefore, increasing tolerance to drought stresses is an important target for improving the performance of soybean in the field [9,10].

The traditional method for breeding drought-tolerant genotypes has been established based on observations of phenotypes, including the wilting degree and/or yield losses for plants grown under controlled conditions or natural drought conditions. However, heavy investment and a long research period are the limitations of conventional breeding for drought-resistant varieties. Moreover, as drought tolerance is a multigenic and quantitative trait, some difficulties arise when attempting to breed for tolerance using conventional approaches. Marker-assisted selection (MAS) for target quantitative traits is an effective approach to utilize the natural variation associated with the drought tolerance of soybean. Up to now, many quantitative trait locus (QTLs) mapping and genome-wide association study (GWAS) efforts have been conducted to identify genomic regions associated with soybean’s drought tolerance. However, the studies mainly focused on traits referring to yields [11,12], fibrous roots [13] and leave-related traits [12,14,15,16,17,18] at the seedling or mature periods under drought stress conditions. Although drought could occur at different stages of soybean growth, seed germination and seedling emergence are potentially considered the most critical stages susceptible to water stress and are pivotal steps for crop propagation [19]. Some studies report several physiological characteristics (including seed germination and seedling growth) as indicators of drought tolerance in specific crop genotypes [20,21]. Thus far, only Liu et al. reported QTLs underlying drought tolerance at soybean’s germination stage through genome-wide association mapping [22]. However, the discovery of new drought-resistant gene resources by map-based cloning has not been reported.

Many transcription factors that regulate genetic pathways under stress include nuclear factor Y (NF-Y), heme-activator protein (HAP) and CCAAT-binding factor (CBF), which is a conserved heterotrimeric complex consisting of NF-YA (HAP2 or CBF-B), NF-YB (HAP3 or CBF-A) and NF-YC (HAP5 or CBF-C) subunits in animal, yeast and plant systems [23,24]. NF-Y proteins are important regulators of stress tolerance in plant growth and development, especially in response to drought stress [23]. Till now, many individual NF-Y subunits have been found in *Arabidopsis*, *Triticum aestivum*, *Zea mays*, *Oryza sativa*, *Glycine max* [25,26,27,28,29,30], for instance, which acts through a previously undescribed mechanismand can improve performance under drought conditions in Arabidopsis [31]. Further, an orthologous maize transcription factor *ZmNFYB2* is involved in drought tolerance as suggested by many stress-related parameters, including stomatal conductance, chlorophyll content, leaf temperature, reduced wilting and maintenance of photosynthesis [32]. Furthermore, the overexpression of *PdNFYB7* promotes drought tolerance and improves water-use efficiency in Arabidopsis, indicating its potential role in breeding drought-tolerant plants that increase production even under water deficiency [25].

The objectives of this study were (i) to elucidate the genetic architecture of drought-response traits at the soybean germination stage using both GWAS and linkage analysis; (ii) to confirm the selected major genetic factors (QTLs) regulating soybean drought tolerance; and (iii) to identify and verify potential candidate genes that might be associated with drought tolerance in soybean.

## 2. Results

### 2.1. The Sensitivity of Soybean Response to Drought Stress Display Significant Variation

We tested four germination-related traits to evaluate the sensitivity of 201 germplasms to drought stress. The results showed that the germination index (GI) as well as the drought-resistance indices of the main root length (MRLI), branch root length (BRLI) and total root length (TRLI) had wide variation among soybean germplasms. The variable coefficient (CV) of GI was the largest, followed by that of MRLI, BRLI and TRLI (Appendix A). It indicates that the above characters respond to drought stress and reflect the sensitivity of different experimental materials to drought.

Of the 201 soybean accessions, two representative soybean cultivars, ‘Maple Arrow’ (tolerant to drought stress) and ‘Hefeng25’ (sensitive to drought stress), was screened as parental lines for constructing a mapping population with a total of 150 recombinant inbred lines (RILs) of F_5:10_ generation, named the MP population. The same traits were tested for the 150 RILs from the MP population. Like that in the soybean germplasm population, the four target traits showed wide variations among RILs (Appendix A). In both the association panel and the RIL population, GI, MRLI, BRLI and TRLI showed continuous variation and normal distribution (Figure 1), with a skew and kurtosis less than one (Appendix A). There were significant positive correlations between each pair of the four traits in the two mapping populations, respectively (Figure 1).

### 2.2. Genotyping of the Association Panel and the MH Population

We identified 20,757 SNPs (MAF ≥ 0.05) from more than 59,000 high-quality SLAF tags from each of the 201 genotypes in the association panel (Appendix A, Appendix A). The SNPs covered 20 soybean chromosomes with the marker density of one SNP per 45 kbp (Appendix A). The mean linkage disequilibrium (LD) across all tested soybean accessions was 169 kbp (Appendix A).

For the linkage mapping population, the whole-genome resequencing depth was >3x for each offspring and >20x for the two parental lines. A total of 5241 bin markers generated from 343,907 high-quality SNPs along the 20 chromosomes were identified based on resequencing (Appendix A). Based on the Kosambi mapping function, a genetic map with a total length of 3694.40 centimorgans (cM) was constructed, covering 20 soybean chromosomes (Linkage groups, LGs). The mean interval between bin markers was 0.70 cM (Appendix A).

### 2.3. Loci and Candidate Genes Associated with the Four Drought-Tolerance Indexes

For GWAS, the fixed and random model circulating probability unification were utilized to identify association signals for the four drought-related traits. In the BLINKalgorithm, a total of 11 QTNs were found to be significantly associated with the four drought tolerance indices (Figure 2A–H, Appendix A), which were distributed in six soybean chromosomes. Among them, one QTN (rs42482818) in chromosome (Chr.) 10 showed pleiotropic and controlled the four targeted traits simultaneously. In addition, the detection results of the MLM model were included in the BLINK model in this study. rs42482818 was detected in the four target features of the MLM model, while it was also the only QTN detected in the MLM model.

By linkage mapping, a total of 22 QTLs were identified related to drought tolerance of soybean at the germination stage. Of them, 4, 5, 7 and 6 QTLs were associated with GI, MRLI, BRLI and TRLI, respectively (Figure 2I, Appendix A).All the QTLs could explain 7.06–16.82% of the phenotypic variance. These 22 QTLs that covered 7 soybean chromosomes represented 9 genomic regions. There were 8 genomic regions corresponding to 21 QTLs showing pleiotropic and controlling 2 or more traits. For instance, the marker interval Chr01_45199493-Chr01_55455938 determined the location of qGI1-1 and qMRLI1-1 and was associated with two traits referring to GI and MRLI; the same case was found for marker intervals Chr07_2699622-Chr07_2535197, Chr14_6008788-Chr14_8536163 and Chr15_15673331-Chr15_17236892, which controlled two traits, simultaneously. The marker intervals Chr07_6349115-Chr07_5410219, Chr16_33378801-Chr16_33722040 and Chr19_45082406-Chr19_50007260 were related to three traits, respectively. Remarkably, the marker interval Chr10_42425169-Chr10_42492796, which controlled all the four drought-related traits, was shared by qGI10-1, qMRLI10-1, qTRLI10-1 and qBRLI10-1. The average genetic contribution rate of this pleiotropic locus was more than 10%, indicating that it is the major QTL for soybean drought tolerance at the germination stage. More importantly, the QTL in Chr.10 was the only locus that was co-located by linkage mapping and GWAS (Appendix A). We speculated that there could be a very important gene for soybean drought resistance in the locus of Chr. 10.

### 2.4. Genetic Feature Analysis for the Candidate Region in Chr. 10

To analyze the sequence diversity of the qDI10-1 region on Chr. 10 among wild soybean and cultivated soybean genomes, we identified 632 SNPs within the 67.6kb genomic region of qDI10-1 that harbored ten candidate genes across a subset of 152 soybean accessions, including 76 lines of *Glycinemax* and 76 lines of *Glycinesoja.* The π value of the qDI10-1 region in wild soybean (3.7 × 10^−3^) was higher than that incultivated soybean (2.3 × 10^−3^). The result suggested that qDI10-1 might be domesticated and selected during the process of soybean domestication from wild type to cultivated soybeans resulting in a decrease in sequence diversity (Figure 3A,B).

We further analyzed the genetic diversity of this QTL region by the sliding window and found that two sub-intervals (42.42–42.43 Mbp; 42.45–42.47 Mbp) were significantly differentiated between wild and cultivated soybeans (Figure 3C). There were five out of the ten candidate genes in the qDI10-1 region located in the two sub-intervals. Of them, *Glyma.10G191700* in the interval of 42.42–42.43 Mbp encodes peroxidase superfamily protein, and *Glyma.10G192000* in the interval of 42.45–42.47 Mbp encodes nuclear factor Y. NF-Y transcription factor played important roles in regulating plant responses to drought stress by increasing antioxidant enzyme activities and osmolyte accumulation, although the peroxidase activity was reported as being enhanced under abiotic and biotic stresses in plants. Hence, we speculated that *Glyma.10G192000* could be the candidate gene for drought tolerance in qDI10-1 of Chr. 10.

### 2.5. The Expression Pattern of NF-Y Transcription Factor of Soybean

In soybean genomes, 29 homologous genes of NF-Y transcription factor were found (Appendix A), in which *Glyma.10G192000* was named as the *GmNFYB17* base in the order of gene identifier. To investigate the response of NF-Y transcription factor homologs to drought stress at the gene expression level in soybean, qRT-PCR was conducted for 29 *GmNFYB* homologous genes in soybean (Appendix A). The results showed that in total 12 genes, including *NFYB*17, showed an early response to drought stress. The significant differential expression could be observed during 0–6h under PEG treatment. The maximum expression level at the early stage of these genes was mostly shown at 4h (Figure 4A). Most of the 29 genes were up-regulated not only at the early stage but also at the late stage of stress (at 48 h). The expression levels of all *NFYB* genes can be induced by drought stress at different times, suggesting their involvement in the drought resistance of soybean. ABA treatment induced the transcription levels of *NFYB* genes (Appendix A), which usually showed increased or decreased expression levels 0–6 h after treatment and finally increased at 12–48 h. This result suggested that most *NFYB* genes were involved in the ABA-dependent signaling pathways in soybean. The tissue-specific expression pattern of *GmNFYB17* and expression induced by drought and ABA showed a consistent increasing expression pattern in soybean (Figure 4B–D).

### 2.6. Overexpression of GmNFYB17 Enhances the Tolerance to Water Deficit in Soybean

Transgenic and non-transgenic plants showed no significant differences in the first four weeks before water was limited. However, seven days after the water-deficit conditions, the non-transgenic plants heavily wilted while the transgenic plants grew well. After 15 days of water deficit, transgenic plants partly progressed to leaf curl while the non-transgenic plants wilted, and their growth was suppressed. Moreover, transgenic plants podded better than non-transgenic seven days post-re-watering (Figure 5A).

The relative water content (RWC) of transgenic plants was higher than non-transgenic plants at any stage in all of the three lines (Figure 5B). In both transgenic and non-transgenic plants, the RWC was higher and peaked at 83.9%, whereas, at the well-watered and re-watered stages, it was significantly lower during drought. The higher RWC of transgenic plants suggested that the expression of *GmNFYB17* contributed to enhancing the capacity for osmotic adjustment in soybean.

The degree of leaf damage in non-transgenic plants showed a vital increase in drought and re-watering, while it was nearly the same in transgenic and non-transgenic plants when they were well-watered (Figure 5C). These results revealed that *GmNFYB17* transgenic plants have stronger drought resistance.

### 2.7. Overexpression of GmNFYB17 Impacts SOD Activity, Proline Content and MDA in Soybean Plant

MDA content increased consistently in both transgenic and non-transgenic plants but was higher in non-transgenic plants (Figure 5D). Higher MDA content is indicative of a greater degree of injury in non-transgenic plants but was indicative of drought tolerance in transgenic plants.

Superoxide dismutase (SOD) activity markedly increased during drought and re-watering. SOD activity of G16 and G18 was twice as much as that of non-transgenic plants after 15 days of the water deficit stress. After re-watering, the activity of SOD in G26 was significantly higher than that of non-transgenic plants (Figure 5E).

Proline contents remained low in both transgenic and non-transgenic plants; however, proline content accumulated sharply in both transgenic and non-transgenic plants 15 days post water stress. Moreover, the proline content is higher in transgenic plants as compared with non-transgenic plants. The proline content in G18 was 586.08 (ug/g) when subjected to drought and was 21.05 (ug/g) in non-transgenic re-watered plants (Figure 5F). Proline accumulation in transgenic leaves indicates the drought tolerance of plants.

### 2.8. Overexpression of GmNFYB17 Impacts Soybean Root Growth

We confirmed the T-DNA insertion and the existence of a single copy of the *GmNFYB17* gene using Southern blot analysis (Appendix A). Moreover, the expression of *GmNFYB17* in all the transgenic lines was much higher than in non-transgenic plants. The maximum level of expression was observed in G16, which was 6.4 times higher as compared with the non-transgenic ones (Appendix A).

We found significant differences in root growth in transgenic and non-transgenic plants. The root system of transgenic plants grew rapidly and demonstrated more branching and lateral roots than non-transgenic roots (Figure 6A). The root lengths of all of the three transgenic lines (G16, G18 and G26) were longer than non-transgenic plants (Figure 6B). The number and length of lateral roots (10 cm) were higher in all transgenic lines, particularly G26, which had the longest roots among all lines. The roots of G26 were almost two times longer than the control group’s roots (Figure 6C), indicating the involvement of the *GmNFYB17* gene in root growth.

The root–top ratio of all three transgenic lines was much higher than the non-transgenic plants and was especially high in G18 and G26, which showed dramatic root growth that doubled the growth in control (Figure 6D).

### 2.9. Overexpression of GmNFYB17 Increases Soybean Yield under Drought Condition

Transgenic and non-transgenic plants showed significant differences in root length, main stem pods, grains per plant, seed diameter and 100-seed weight. Transgenic lines showed higher root length compared with control, and G26 plants showed significantly higher plant height compared with control (Figure 7A,C).

The results of investigating plant and agronomic traits showed that the transgenic plants grew more branches, enhanced the main stem nodes, podsand increased the number of grains per plant; thus, causing higher yield (Figure 7B).

The seed diameter of each transgenic strain was bigger compared to that of non-transgenic plants. The average diameter of seeds in G16, G18 and G26 was 5.86, 6.17 and 5.80 mm, respectively, against 4.61 mm for non-transgenic ones (Figure 7D,F). The 100-seed weight of the transgenic lines was greater than those of non-transgenic plants (Figure 7E). Thus, the overexpression of *GmNFYB17* could also improve the yield traits in soybeans.

## 3. Discussion

Soybean is sensitive to drought stress and improving the tolerance to drought stress, is of great significance for the stable yield of soybean. Thus far, many studies have used linkage mapping [11,12,13] or GWAS [22] to mine soybean drought tolerance QTL and are expected to apply it in MAS. However, both linkage mapping and GWAS have certain limitations, and it is a trend to use the combination of the two to co-localize loci [33]. In terms of soybean drought tolerance, few studies used linkage mapping and GWAS to mine QTL jointly. In this study, both linkage mapping and GWAS were used to mine drought tolerance QTLs, 22 QTLs and 11 SNPs were detected separately (Appendix A); these loci lay the foundation for MAS, and one locus, qGI10-1, was detected by two methodsjointly, exhibiting higher accuracy for the mining of drought tolerance genes.

Although there had been many advances in QTL mining for drought resistance in soybean, there was still no report on the discovery of new drought resistance gene resources through map-based cloning. The qGI10-1 detected in this study had a differentiated region between wild soybean and cultivated soybean. The nuclear factor Y transcription factor *GmNFYB17* was detected in the differentiated region of qGI10-1. Our data suggested that soybean *GmNFYB17* is significant for drought stress tolerance, and its overexpression in soybean improved drought tolerance and accelerated root growth. ABA, salt stress and drought regulate *GmNFYB17* suggesting its role in stress responses.

Drought stress damages the cellular membranes and macromolecules and simultaneously causes the production of reactive oxygen species (ROS) and other toxic substances [34,35]. Plants remove these toxic compounds with their antioxidants, thereby enhancing the antioxidant defense levels; thus, drought tolerance contributes to stress resistance [36,37]. In our study, *GmNFYB17* transgenic lines showed higher SOD concentration compared with control (plants without overexpressed *GmNFYB17*) to help scavenge the oxygen free radicals. Therefore, the overexpression of *GmNFYB17* may enhance tolerance to drought stress response.

Previous studies showed that proline is the major osmolyte that contributes to osmotic adjustment and enhancement of stress tolerance in plants under osmotic stress conditions [36]. Moreover, hydrophilic proline has a strong, stable colloidal protoplasm that lowers the freezing point and prevents cell dehydration [38]. The accumulation of proline in barley positively enhances drought resistance [39,40,41]. The present results infer that the accumulation of proline in *GmNFYB17* transgenic lines could be an indicator of drought stress. Hence, *GmNFYB17* might improve water-limited tolerance by increased proline accumulation.

Enhanced root growth and a well-developed root system can improve the drought tolerance of plants [23,25]. Our studies provided support for the role of *GmNFYB17* in regulating root length and lateral root growth. Increased lateral root production and root elongation were relevant to overexpression of *GmNFYB17*; therefore, both the root–top ratio and the root surface area increased. Thus, the overexpression of *GmNFYB17* in transgenic soybean contributed to enhance drought resistance via the development of root systems.

Drought is a major environmental constraint responsible for grain production and crop yield. Agronomic traits can represent the characteristics of various crops whose related traits are regarding the crop growth period, plant height, leaf area, fruit weight and so on. Soybean yield is determined by seed size and weight, and in our study, both factors were higher in *35S:GmNFYB17* transgenic lines. Therefore, our study suggests that the *GmNFYB17* gene may be associated with yield traits through gene regulation and could be beneficial for the development of higher-yielding varieties. Therefore, *GmNFYB17* could play a critical role in a high yield of soybeans.

## 4. Materials and Methods

### 4.1. Mapping Population

Two mapping populations were used in this study. To construct the phenotypic diversity association panel, 201 soybean accessions, including 39 landraces and 162 elite cultivars, were collected worldwide. Of these, the 179 Chinese accessions originated from eight different provinces and three sub-ecological areas within the two main regions of soybean cultivation in China (latitude: 53–39° N). These accessions were selected from the 20,000 core Chinese germplasms. We also analyzed 21 soybean cultivars that originated from outside of China (Appendix A).

From the 201 analyzed germplasms, cultivar ‘Maple Arrow’, which was tolerant to drought, and the cultivar ‘Hefeng25’, which was sensitive to drought, were used as parental lines to construct a recombinant inbred population with 150 lines. The F_5:10_ generation of the MH population was used for the linkage mapping of soybean tolerance to drought at the germination stage.

### 4.2. Phenotyping

The PEG6000 with a concentration of 15% was used to assess the drought tolerance of the seeds of each line from the association panel and MH population. Briefly, thirtyhealthy seeds from each line were selected. The soybean seeds were sterilized for 16 h in a desiccator containing 6 mL of HCl (38%) and 96 mL of NaClO (8%). Sterilized seeds were put into Petri dishes with a diameter of 9 cm with a piece of filter paper in each dish. A total of 20 mL of PEG6000 (15%) or water as control was added to the dishes. The dishes were incubated in the dark in a germination chamber at 25 ± 1 °C, and the supplement was added to each dish in 20mL solution every two days. The number of germinated seeds was counted at 3d, 6d, 9d and 12d after incubation and the germination index (GI) was calculated by the following formula:GI = DGI_ij_/NDGI_ij_

DGI_ij_ and NDGI_ij_ are the germination index for the jth trait of ith tested line under drought condition and normal condition, respectively. DGI_ij_ and NDGI_ij_ were calculated with the number of germination seeds at 3d, 6d, 9d and 12d after treatment or control by the formula:GI_ij_ = (1.00) × nd3 + (0.75) × nd6 + (0.5) × nd9 + (0.25) × nd12

The nd3, nd6, nd9 and nd12 was the number of germinated seed under drought condition or normal condition. Otherwise, the main root length (MRL), branch root length (BRL) and total root length (TRL) were measured at 12d after incubation. For each tested line, the drought tolerance index of the three traits, namely MRLI, BRLI and TRLI, was calculated by LD/LN. Where the LD was root length under drought treatment, and LN was root length under normal conditions of seed germination. The study was carried out in a greenhouse (constant temperature 26 °C, light–dark cycle 16 h/8 h) at Northeast Agricultural University with 3 consecutive repeated batches in May and June. The phenotypic results were averaged from 3 batches.

### 4.3. SLAF-Seqand SNP Calling for the Association Panel

Genomic DNA samples from each accession in the association panel were obtained from fresh leaves using the methods of Sunet al. [42]. SLAF-seq was used to analyze these genomic DNA samples [42]. A double enzyme system with *Mse*I and *Hae*III (Thermo Fisher Scientific Inc., Waltham, MA, USA) was used to digest the genomic DNA of each accession. We obtained more than 50,000 sequencing tags, each 300–500 bp long, and used these to construct sequencing libraries. These sequencing tags were evenly distributed across the unique regions of the 20 soybean chromosomes. The 45 bp sequence read at both ends of the fragment in each library was generated by Illumina Genome Analyzer II (Illumina Inc., San Diego, CA, USA); barcodes were used to identify each sample.

The raw paired-end reads were mapped onto the reference genome (assembly Glycine_max_v2.1) [43] using Short Oligonucleotide Alignment Program 2 (SOAP2) [44] (http://soap.genomics.org.cn; accessed on 8 March 2020). When multiple reads were mapped to the same genomic position, a SLAF group was defined. If the genomic DNA of an accession is not fully digested by the double enzyme system, some reads mapped to the reference genome might overlap with more than one SLAF tag [42]. In these cases, these reads were linked with both SLAF tags for that accession. A threshold of MAF ≥0.05 was used for SNP calling. Genotypes were regarded as heterozygous when the ratio of minor allele depth to total sample depth was ≥1:3.

### 4.4. Genomic Resequencing of the RIL Population

Genomic DNA for the MH population and the parental lines were prepared as described by Qi et al. [45]. Sequencing libraries for these samples were constructed and sequenced on anIllumina HiSeq2500 sequencing platform following the manufacturer’s instructions. The sequencing reads for the RI population, and the parental lines were aligned to the soybean reference genome (assembly Glycine_max_v2.1) [43] using Short Oligonucleotide Alignment Program 2 (SOAP2) [44]. GATK 4 [46] (https://gatk.broadinstitute.org/hc/en-us; accessed on 2 April 2020) was used to identify polymorphic SNPs between the RI population and the parental lines. Co-segregating SNPs were separated into bins, and a bin map was constructed based on the recombinant breakpoints of the MH population with HighMap [47] (http://highmap.biomarker.com.cn/; accessed on 5 April 2020).

### 4.5. LD Pattern Analysis of the Association Panel

For each pair of SNPs where the MAF of each SNP was ≥0.05 and the integrity of each SNP was ≥50%, the LD (R^2^) was determined using Haploview v4.2 [48] (https://www.broadinstitute.org/haploview/haploview; accessed on 20 April 2020).

### 4.6. Identification of QTLs/QTNs for Drought Tolerance of Soybean

To identify the loci associated with soybean drought tolerance, we performed a GWAS using the BLINK algorithm in the GAPIT 3 package [49] (http://zzlab.net/GAPIT; accessed on 5 May 2021), based on the SLAF-Seq-identified SNPs in the 201 soybean accessions. The MLM (K + PCA) model in the Tassel 5.0 [50] (http://sourceforge.net/projects/tassel; accessed on 10 May 2020) was applied to compare the results obtained with the BLINK. The *p*-value ≤ 1.74 × 10^−6^, which was set as the threshold to declare whether significant association signals existed.

IciMapping v4.1 [51] (http://www.isbreeding.net/; accessed on 15 May 2020) was used to map QTLs controlling soybean drought tolerance. The threshold value was set to 2.5; the 99% confidence intervals for the identified QTLs were subtracted following Qi et al. [45].

### 4.7. Genetic Diversity Analysis of the Genomic Region qDI10-1

To investigate whether the genomic region of qDI10-1 is related to soybean domestication from wild soybean to cultivated soybean, the genetic diversity, Tajima’s D and *F_ST_* were evaluated using the polymorphic SNPs of 152 soybean accessions, including 76 wild soybeans and 76 cultivated soybeans. The SNP data set was generated from genome resequencing reported through the project of large-scale sequencing of germplasms to develop genomic resources for soybean improvement (https://www.soybase.org/; accessed on 28 May 2020).

### 4.8. Soybean Plant Culture and Stress Treatments

Soybean seeds (Maple Arrow) were grown in soil containing vermiculite (soil/vermiculite = 1:1) in a growth chamber. The growth chamber was maintained at 25 °C, with 65% relative humidity under 16h light and 8h dark cycles. For gene expression assay, 20-d-old seedlings were transferred to MS medium supplemented with water (control condition). After three days of recovery, stress treatments were conducted by adding chemicals to the MS medium with 8% PEG6000, 200 mM NaCl or ABA. The leaves samples from treatments and control were collected at 0, 2, 6, 12, 18, 24 and 48 h post-treatment. Likewise, leaves harvested from control conditions were collected. All of them were then immediately frozen in liquid nitrogen and stored at −80 °C until RNA extraction.

### 4.9. qRT-PCR Assay

The total RNA was extracted by RNAiso Plus (TaKaRa, Kyoto, Japan, 9108). The first-strand cDNA was synthesized using TIANScript RT Kits (KR104; Tiangen, Beijing, China) following the manufacturer’s instructions. qRT-PCR was performed by using an ABI-7500 fast platform with the TB Green^®^ Fast qPCR Mix (TaKaRa, Kyoto, Japan, RR430A). The program was run under the following settings: pre-denaturation at 95 °C for 30 s, followed by a 40-cycle program (95 °C, 5 s; 60 °C, 34 s; per cycle).The soybean housekeeping gene *GmActin4* (GenBank accession no. AF049106) was used as the internal reference gene. The gene expression rate was calculated by the 2^−ΔΔCT^ method. All experiments were analyzed with three technical and three biological replications. Primer sequences were listed in Appendix A.

### 4.10. Cloning of GmNFYB17 Genes and Construction of Plant Expression Vector

A full-length cDNA of *GmNFYB17* was cloned using primers (forward primer: 5′GGT CTA GAC AAA GGT GCA TTG GTG GTC3′; reverse primer: 5′ATG AGC TCC GTA CAA GCA TTC AAG GGA3′). The forward and reverse primers included *Hind* III and *EcoR*I digestion sites. The recombinant vector *pCAMBIA-3300-GmNFYB17* was transferred into the *Agrobacterium tumefaciens* strain LBA4404.

### 4.11. Plant Transformation and Southern Blot Analysis

The cotyledonary explant transformation method was used in the experiment. The *GmNF-YB17* gene was transformed into the cultivar ‘DN50’ seedlings by *Agrobacterium*.

Total DNA was isolated from the non-transgenic and different transgenic lines plants using the cetyltrimethyl ammonium bromide (CTAB) protocol. The ORF of *GmNFYB17* was labeled using PCR DIG Labeling Mixplus for Southern analyses (Roche, Mannheim, Germany). Genomic DNA (2 μg) from each sample was digested with five units of *Hind* III and *EcoR*I, fractionated on a 0.8% (*w*/*v*) agarose gel, denatured with NaOH and transferred onto ZetaProbe GT nylon membrane (Bio-Rad, Hercules, CA, USA). The probe labeling, hybridization, washing and detection were performed following the manufacturer’s instructions (DIG High Prime DNA Labeling and Detection Starter Kit).

### 4.12. Transgenic Plant Material and Growth Conditions

Three populations of T_3_ generation of *GmNFYB17* transgenics derived by selfing were selected as experiment materials, and they were named G16, G18 and G26 in this paper. The receptor ‘DongNong 50 (DN50)’ was used as a control. The plants were grown in pots under natural light conditions at an average temperature of 28/16 °C (day/night) in a greenhouse.

### 4.13. Identification of Root Growth and Root-Top Ratio

Seeds of the transgenic and non-transgenic soybean were planted in different plastic bags in a greenhouse under the same growing conditions as above. Six-week-old roots (or when they reached the bottom of the bag) were subject to qRT-PCR. At the same time, taproot length and the number of lateral roots were measured. Underground and overground portions of the plant parts were placed in different pre-weighed vials. Samples were then oven-dried at 105 °C for 10 min and at 80 °C for 12 h and weighed to determine the dry weight of underground (DW_1_) and overground portions (DW_2_). The calculation formula was
Root-top ratio (%) = (DW_1_)/(DW_2_) × 100

### 4.14. Response of Transgenic Plants under Water Deficit Stress

Both transgenic and non-transgenic plants were grown in different pots. The early growth period was divided into three stages related to the water treatment. Stage one was the initial four weeks of normal watering, where in the soil’s relative moisture content in buckets was adjusted to a saturation of 40%. The second stage was the following two weeks when watering was withheld, and the soil’s relative moisture content dropped by 20% at the end of the drought stress treatment. The third stage was the last week when the soil’s relative moisture content increased up to 45% by re-watering. Samples were taken for experiment analysis at the three stages, which were called well-watered, drought and re-watered, respectively. The plants grew further to allow seed set.

### 4.15. The Determination of Leaf Relative Water Content (RWC)

The leaves of each sample were placed in a pre-weighed flask, and the flasks with samples were weighed to obtain the weight (W) of the leaves. The samples were immediately hydrated to full turgidity for 10 h under normal room light and temperature. Then, the samples were taken out of the water and dried on filter paper before weighing to obtain the fully turgid weight (TW). Samples were then oven-dried at 105 °C for 10 min and at 80 °C for 12 h and weighed to determine the dry weight (DW). The calculation formula used was
RWC (%) = [(W − DW)/(TW − DW)] × 100

### 4.16. Determination of Physiological and Biochemical Indicators

#### 4.16.1. Superoxide Dismutase (SOD) Measurement

To provide an assessment of the qualitative difference in activities of the various SODs in transgenic and control leaves, proteins were extracted from three biological replicates (individual plants) in three replicate experiments and run on native protein gels. The leaves of each sample were well homogenized with 5 mL of ice-cold 0.05 M/L phosphate buffer (pH 7.8) and centrifuged at 12,000× *g* for 15 min at 4 °C. A total of 5 mL of supernatant was collected for SOD assays. The reaction mixture contained 100 µL crude enzyme extract, 2.5 mL [13 µM/L L-methionine (Met)], 0.25 mL [63 µM/L nitrotetrazolium blue chloride (NBT)], 0.15 mL [13 µM/L lactoflavin] and 0.05 mL [13 µM/L phosphate buffer (pH 7.8)]. The reaction started by adding lactoflavin, and the absorbance was measured at 560 nm by using a Bio-Rad UV/VIS spectrophotometer after 20 min of incubation at 24 °C under continuous light (4000 lx, irradiated from a fluorescent lamp). One unit of SOD was defined as the amount of enzyme-producing 50% inhibition of NBT reduction under the assay condition. SOD activity was expressed as units/mg protein^−1^min^−1^. The experiment was performed three times.

#### 4.16.2. Proline Measurement

A total of 1 g of leaves of each sample was homogenized in 4 mL of sulfosalicylic acid 3%, and the homogeneous mixture was centrifuged at 13,000 rpm for 15 min at 4 °C. The mixture was homogenized and extracted in a boiling water bath for 10 min. After cooling down to room temperature, the homogenates were refrigerated at 4 °C and then centrifuged at 3000× *g* for 10 min, and 5 mL of each supernatant was collected for proline assays. A total of 2 mL of the extracting solution was combined with 2 mL of acid-ninhydrin reagent and glacial acetic acid in a test tube and heated in a water bath maintained at 100 °C for 50 min. The reaction was terminated in an ice bath until room temperature. The reaction mixture was then extracted with 4 mL of toluene. Finally, the 4ml of the toluene phase was removed for absorbance measurement at 520 nm in a DU640 spectrophotometer. Toluene was used as the blank control. The content of proline was measured by the same method as described above for making a standard curve.

#### 4.16.3. Malondialdehyde (MDA) Measurement

MDA content was determined using the thiobarbituric acid (TBA) reaction producing the reddish brown bilatriene under acidic and high-temperature conditions. For each treatment, leaf samples were grounded in 5 mL of 0.1% (*w*/*v*) trichloroacetic acid (TCA), mixed with 5 mL of 0.5% (*w*/*v*) TBA, and centrifuged at 10,000× *g* for 10 min at 4°C. The supernatant (5 mL) was boiled for 30 min, cooled to room temperature and centrifuged at 4000× *g* for 10 min. The clear supernatant was analyzed by monitoring the difference in absorbance at 450, 532 and 600 nm. Each sample was repeated three times.

#### 4.16.4. Leaf Damage Determination

Leaves of each treatment were placed in a flask with 20 mL of deionized water and treated under vacuum for 10 min. Then, the conductivity (S1) of samples was determined. The samples were then boiled for 10 min and cooled down to room temperature. The conductivity was determined again as (S2). The evaluation of leaf damage (%) was calculated as
(L) = S1/S2 × 100

#### 4.16.5. Investigation of Plant and Agronomic Traits

When the transgenic and non-transgenic plants were grown to seed set, agronomic traits including plant height, root length, branches, main stem nodes, main stem pods grains per plant, seeds diameter and 100-seed weight were investigated.

## 5. Conclusions

This study identified 11 SNPs and 22 QTLs significantly associated with soybean drought tolerance through GWAS and linkage mapping. One of these loci, qGI10-1, was mapped in both methods. Using the difference in the qGI10-1 interval between wild soybean and cultivated soybean, the nuclear factor Y transcription factor *GmNFYB17* was selected as a candidate gene for the study. The overexpression of *GmNFYB17* in soybean enhanced drought resistance and yield accumulation, demonstrating that the *GmNFYB17* gene positively regulates drought resistance and root growth in soybean. These results provide important insights into the MAS and molecular mechanisms of soybean drought tolerance.

## Figures and Tables

**Figure 1 ijms-23-07242-f001:**
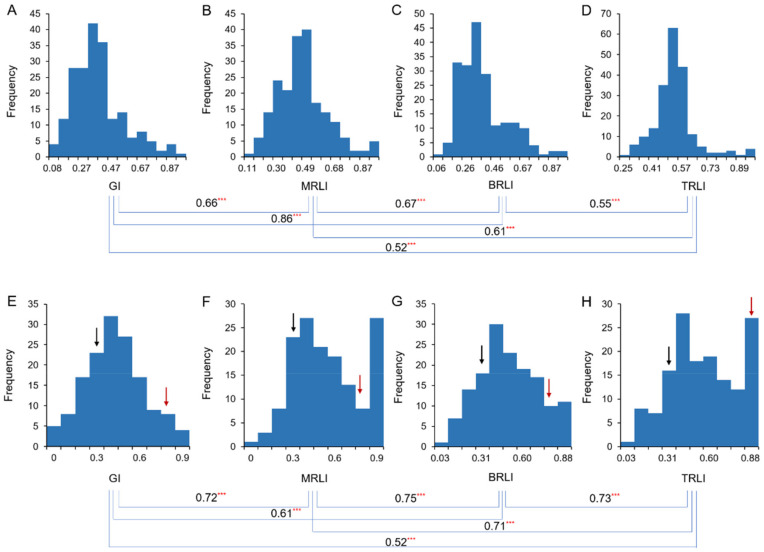
Distribution of drought indices for four drought-related traits at the germination stage in soybean. ***: *p*-value ≤ 0.001. (**A**–**D**) 201 soybean accessions. (**E**–**H**) MH population. Black arrows indicate ‘Hefeng25’, and red arrows indicate ‘Maple Arrow’ in each figure.

**Figure 2 ijms-23-07242-f002:**
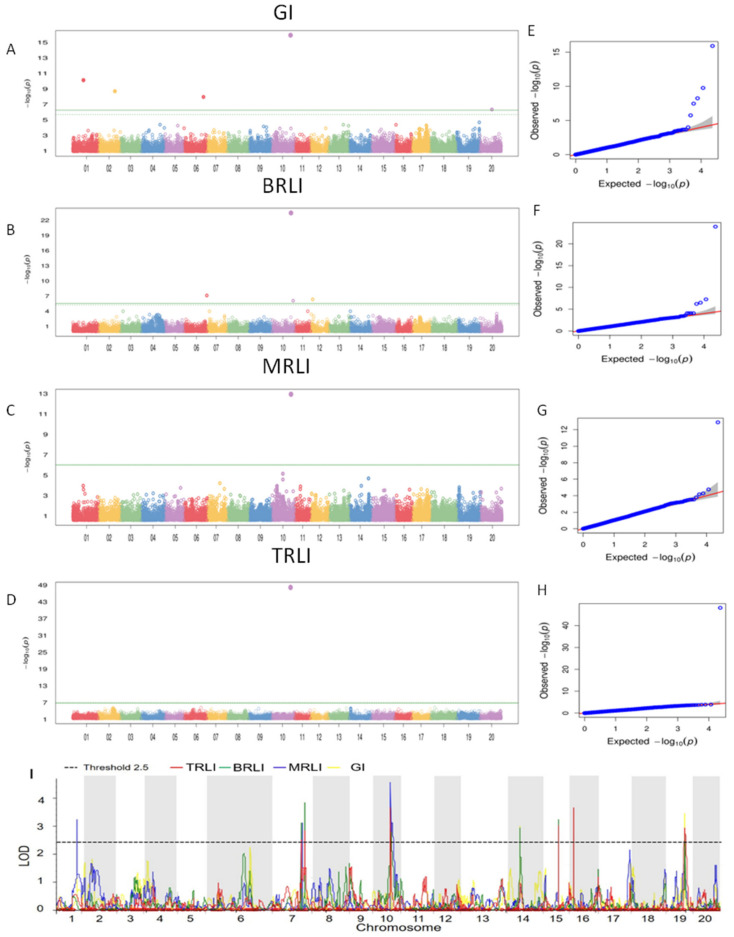
GWAS and QTL mapping of soybean tolerance to drought at the germination stage. (**A**–**D**) Manhattan plot of the four traits. (**E**–**H**) QQ plot of the four traits. (**I**) The QTL mapping results of soybean tolerance to drought at the germination stage.

**Figure 3 ijms-23-07242-f003:**
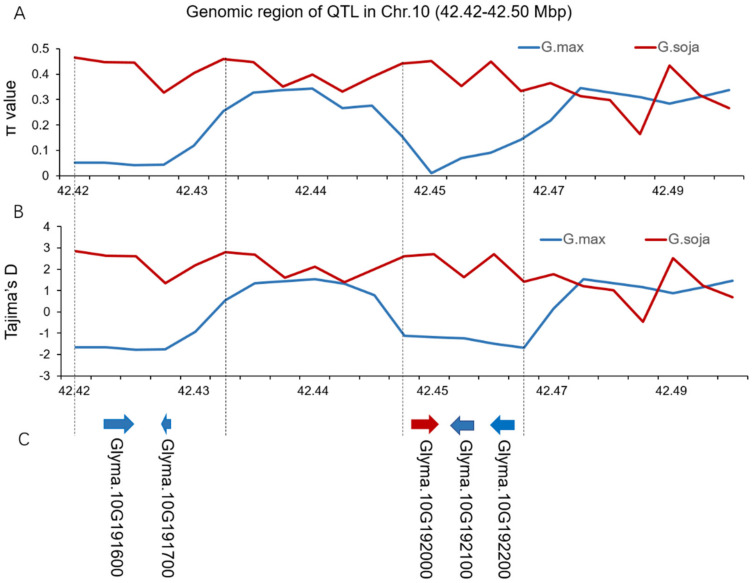
Genomic diversity of the qGI10 region. (**A**) Genetic diversity (π) of Glycine soja (the green line) and Glycine max (the blue line) in the qGI10 region; (**B**) Tajima’s D value of Glycine soja (the green line) and Glycine max (the blue line) in the qGI10 region; (**C**) Candidate genes in qGI10 region.

**Figure 4 ijms-23-07242-f004:**
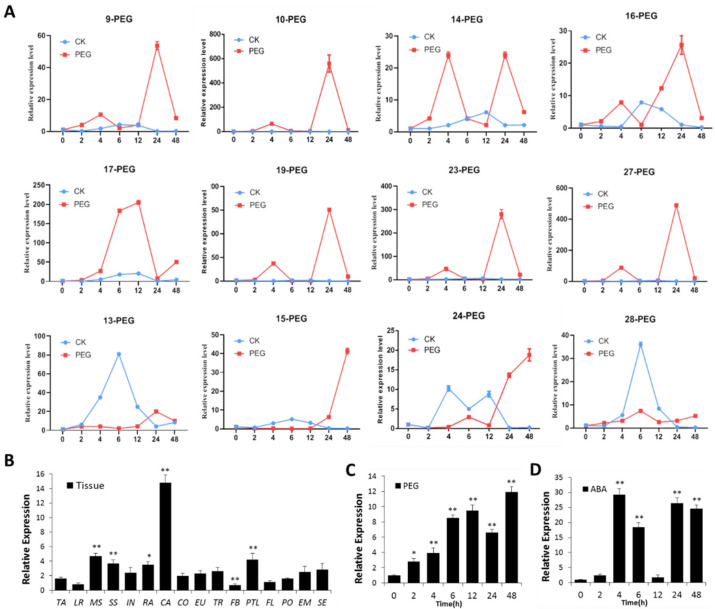
Expression patterns of *GmNF-YB17* in soybean. *: *p*-value ≤ 0.05; **: *p*-value ≤ 0.01; (**A**) Twelve *NFYB* genes’ early response to drought stress, in which *NFYB*13, 15, 24 and 28 are down-regulated; (**B**) Tissue-specific expression of *GmNF-YB17* in soybean Maple Arrow. Tissues tested include taproot (TA), lateral root (LR), main stem (MS), side stem (SS), internode (IN), radicle (RA), caulicle (CA), cotyledon (CO), euphylla (EU), trifoliate leaf (TR), flower bud (FB), stem of trifoliate leaf (STL), flower (FL), pod (PO), embryo (EM) and seed (SE); (**C**,**D**) Expression levels of *GmNF-YB17* in leaves of soybean treated with 100 μmol ABA and 8% PEG6000.

**Figure 5 ijms-23-07242-f005:**
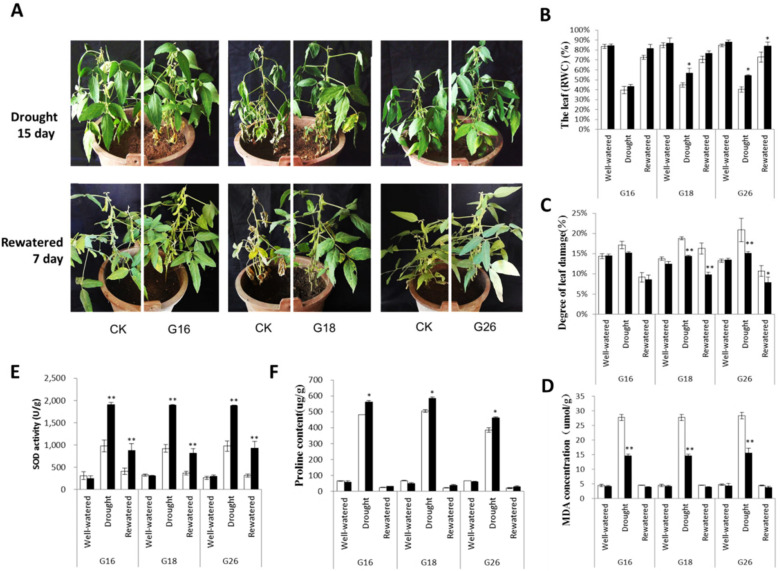
The *GmNFYB17* transgenic soybean lines under drought treatment. (**A**) Morphology of transgenic and non-transgenic plants under drought conditions. Water was withheld for 15 d, and then plants were re-watered for 7 d. G16, G18 and G26 are transgenic lines; CK is soybean DN50; (**B**,**C**)The leaf relative water content (RWC) and leaf damage of transgenic lines. G16, G18, G26 and non-transgenic control (CK) during the well-watered, drought and re-watered stage; (**D**–**F**) Comparison of physiological and biochemical indicators (MDA, SOD, Proline) between transgenic and non-transgenic plants. *: *p*-value ≤ 0.05; **: *p*-value ≤ 0.01.

**Figure 6 ijms-23-07242-f006:**
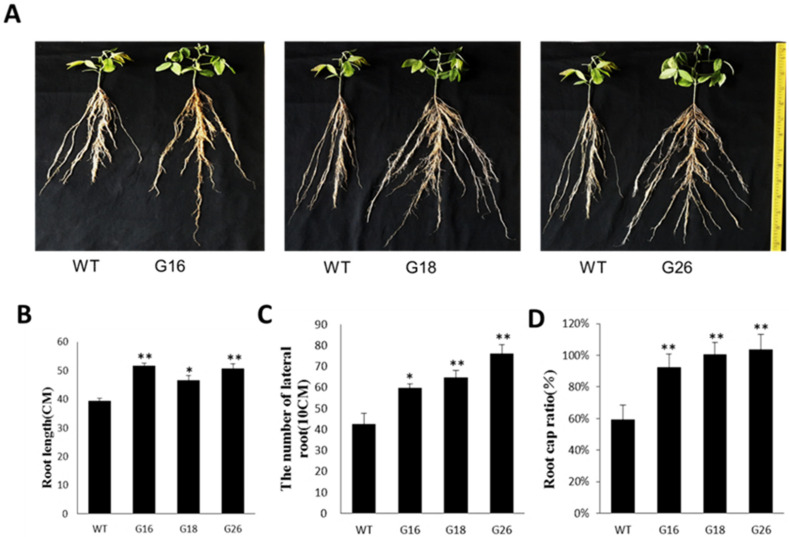
Identification of root growth and root–top ratio of the transgenic soybean. Seeds of three transgenic lines and non-transgenic control plants were planted in different plastic bags in a greenhouse under normal conditions for 6 weeks. *: *p*-value ≤ 0.05; **: *p*-value ≤ 0.01; G16, G18, G26: transgenic lines; CK: non-transgenic control plants. (**A**) Morphological differences in the primary roots of 6-w-old seedlings; (**B**) The difference in root length between three transgenic lines and non-transgenic seedlings; (**C**) The number of lateral roots (10cm) of 6-w-old seedlings; (**D**) The difference in root–top ratio among G16, G18, G26 and CK.

**Figure 7 ijms-23-07242-f007:**
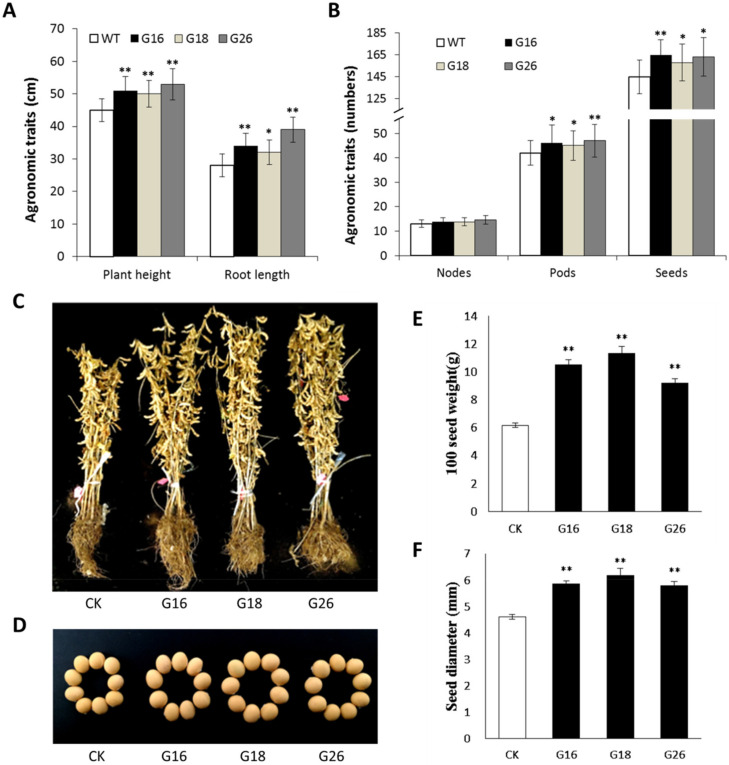
Investigation of plant and agronomic traits of transgenic lines. *: *p*-value ≤ 0.05; **: *p*-value ≤ 0.01; (**A**) Plant height and root length of G16, G18, G26 and CK; (**B**) The number of branches, nodes, pods and seeds in the transgenic lines and non-transgenic plants; (**C**) The phenotype of *GmNFYB17* after harvest; (**D**) Comparison of seed size among *GmNFYB17* lines and CK; (**E**) The 100-seed weight of the transgenic lines and non-transgenic plants; (**F**) The diameter of transgenic and non-transgenic seeds.

## Data Availability

Not applicable.

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
