# Peer review of "A Nuclear Factor Y-B Transcription Factor, GmNFYB17, Regulates Resistance to Drought Stress in Soybean"

_ijms, 2022, doi:10.3390/ijms23137242_

Round 1

Reviewer 1 Report

Comments:

The MS entitled, “A Nuclear FactorY-B transcription factor, GmNFYB17, regulates resistance to drought stress in soybean” by Sun et al. described the role of NFYB17 gene from Glycine max in drought stress. The work executed well and represented in good format. The following comments should be fulfilled to improve the MS.

1.      Expand GWAS in page no.2

2.      In 2.2. paragraph better to give the percentage details of HCL and sodium hypochlorite solution

3.      In 2.7. paragraph, line no 6. What do you mean aqueous? And rearrange this sentence

4.      In 2.7. paragraph, line no 5 remove the “by” and should be reported through

5.      In 2.13. paragraph, line no 3, remove the were samples

6.      Provide italics to genus and species name. Ex: Paragraph 3.4

7.      A short conclusion and perspectives would be effective for the readers

8.      Authors are advised to check the typo errors throughout the MS

9.      Why don’t author can try to represent the expression pattern in heatmap for Fig.S4.

Reviewer 2 Report

In this study, authors performed GWAS and QTL mapping to identify many significant genomic region associated drought tolerance in soybean. As a result, authors successfully identified qGI10-1 loci association with the phenotype. Further, identified the candidate gene GmNFYB17 within the qGI10-1. I found this article is very interesting since authors designed the experiments well and confirmed candidate gene through functional study in association with drought tolerance. The result presented here will be a valuable resource for future genetic investigations.

However, this study has some minor flaws that are a minor cause of concerns.

Page/Line –Edit/Comment

1.     Why authors chose only FarmCPU model for GWAS study? I think MLM model is more suitable to identify the significant genomic region with the phenotype if you are working with 20,757 SNPs since it covers both Q and K matrix.

2.     I would suggest BLINK model instead of FarmCPU since BLINK considered being a latest model with Bayesian information and LD. Since BLINK improves, statistical power compared to FarmCPU.

3.     It is always better to perform the GWAS with multi-models because you can choose more reliable significant SNPS across the models

4.     How many replication you have performed for phenotype and performed the GWAS study? Generally, GWAS study should be performed with 2-3 seasons/replication to choose most reliable significant SNPs.  

5.     Please elaborate your discussion part since I found less informative.
